# LANGUAGE MODELS AS BLACK-BOX OPTIMIZERS FOR VISION-LANGUAGE MODELS

## ABSTRACT

Vision-language models (VLMs) pre-trained on web-scale datasets have demonstrated remarkable capabilities across a variety of vision and multimodal tasks. Currently, fine-tuning methods for VLMs mainly operate in a *white-box* setting, requiring access to model parameters for backpropagation. However, many VLMs rely on proprietary data and are not open-source, which restricts the use of white-box approaches for fine-tuning. Given that popular private large language models (LLMs) like ChatGPT still offer a language-based user interface, we aim to develop a novel fine-tuning approach for VLMs through *natural language prompts*, thereby avoiding the need to access model parameters, feature embeddings, or output logits. In this setup, we propose employing *chat-based LLMs as black-box optimizers* to search for the best text prompt on the illustrative task of few-shot image classification using CLIP. Specifically, we adopt an automatic "hill-climbing" procedure that converges on an effective prompt by evaluating the accuracy of current prompts and asking LLMs to refine them based on textual feedback, all within a conversational process without human-in-the-loop. In a challenging 1-shot learning setup, our simple approach surpasses the white-box continuous prompting method (CoOp) by an average of $1.5\%$ across 11 datasets including ImageNet. Our approach also outperforms OpenAI's manually crafted prompts. Additionally, we highlight the advantage of *conversational feedback* that incorporates both positive and negative prompts, suggesting that LLMs can utilize the implicit "gradient" direction in textual feedback for a more efficient search. Lastly, we find that the text prompts generated through our strategy are not only more interpretable but also transfer well across different CLIP architectures in a black-box manner.

## 1 INTRODUCTION

Vision-language models (Radford et al., 2021; Alayrac et al., 2022; Wang et al., 2022; Li et al., 2023) (VLMs) excel at a wide range of classic vision and multimodal (Deng et al., 2009; Lin et al., 2014; Young et al., 2014; Antol et al., 2015; Goyal et al., 2017) tasks, surpassing the performance of their fully-supervised counterparts on downstream tasks even when fine-tuned with minimal data (Lin et al., 2023; Zhou et al., 2022a). However, fine-tuning VLMs typically requires transparent "white-box" access to the model weights, such as gradient-based approaches that rely on backpropagation.

**VLMs as black-box services.** Despite community efforts to collect web-scale public datasets (Schuhmann et al., 2021; 2022) and to replicate proprietary models (Ilharco et al., 2021), an increasing number of VLMs (Alayrac et al., 2022; Yu et al., 2022; OpenAI, 2023; Wang et al., 2022; Driess et al., 2023) are not releasing their weights due to privacy and legal concerns (Li et al., 2020; Madry et al., 2017). Given that contemporary black-box models like ChatGPT (Ouyang et al., 2022; OpenAI, 2023) and DALLE (Ramesh et al., 2022) still offer a language-based user interface, we anticipate that future *black-box* VLMs may be accessed exclusively through APIs that facilitate input and output in *natural language*. Consequently, one cannot use traditional *white-box* fine-tuning strategies that rely on model weights, feature embeddings, and output logits.

**Manual prompting.** Manual prompt engineering has been proven successful in adapting black-box LLMs to language tasks (Wei et al., 2022; Kojima et al., 2022). Similarly, carefully crafted prompts can enhance the performance of VLMs. For instance, CLIP has demonstrated improved zero-shot recognition performance using specifically tailored prompts, such as `"a photo of a class"`

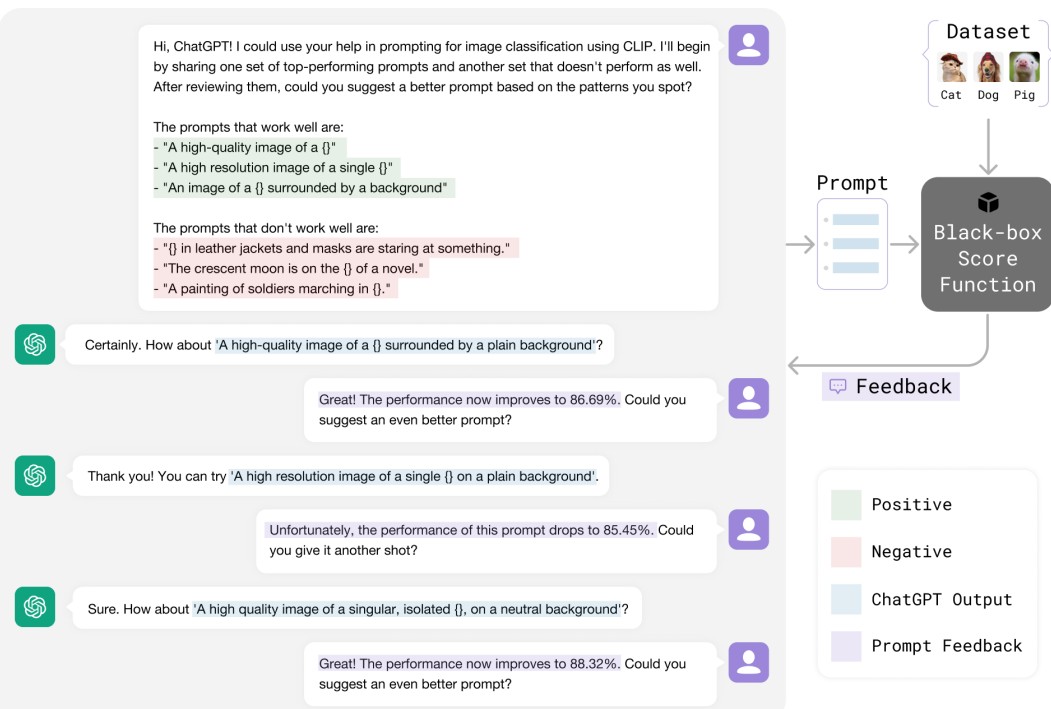

Figure 1: **Conversational prompting for black-box VLMs.** Similar to how human prompt engineers iteratively test and refine prompts, we employ ChatGPT (OpenAI, 2023; Ouyang et al., 2022) to continuously optimize prompts for black-box vision-language models (VLMs). Our iterative approach assesses the performance of ChatGPT-generated prompts on a few-shot dataset (highlighted in blue) and provides feedback (marked in violet) to ChatGPT through simple conversations, as depicted in the illustrative figure. This straightforward method delivers state-of-the-art results for one-shot image classification across 11 datasets using CLIP, operated in a black-box manner without accessing model weights, feature embeddings, or output logits. We show that providing both positive (in green) and negative prompts (in red) enhances efficiency. Remarkably, our approach outperforms both white-box methods such as gradient-based continuous prompting (CoOp (Zhou et al., 2022a)) and human-engineered prompts (Radford et al., 2021; Menon & Vondrick, 2022) in this extremely low-shot scenario. This figure only shows a typical conversation using ChatGPT's web user interface. Our code implementation follows this pattern using the ChatGPT API. We detail and ablate the prompts in section 7.

for Internet photos and `"a satellite image of a class"` for satellite imagery. Despite its effectiveness, manual prompting can be a laborious process, inspiring efforts to explore automated prompt creation and thereby remove the need for human involvement. These strategies typically leverage a LLM as a knowledge base to create rich visual descriptors that augment the prompts for each class (Menon & Vondrick, 2022; Pratt et al., 2022) in a zero-shot fashion.

**Human-free prompting with conversational LLMs (our approach).** We show how to effectively leverage chat-based LLMs (OpenAI, 2023) to emulate human-level prompt engineering *without* any human input. We address an illustrative low-shot image classification task, aiming to find the best class-agnostic prompt (or "template") for image classification with CLIP. We start with a random set of prompts and evaluate the one-shot training accuracy of each. Then, akin to human prompt engineering, our method repeatedly presents ChatGPT with the best and worst prompts, asking it to review the results and suggest an improvement (see Figure 1). Our approach can be implemented through simple chatting with LLMs, leading us to dub our method *conversational prompting*.

**Learning with implicit "gradients" provided through conversational feedback.** One of our key findings is that LLMs can learn the difference between effective and ineffective prompts, and can use this implicit "gradient" direction provided through language to perform more efficient searches. Compared to previous automatic prompting methods that only use LLMs as a knowledge base (Menon

& Vondrick, 2022; Pratt et al., 2022) or paraphrasing tool (Zhou et al., 2022b), we show a novel use of LLMs as an *optimizier* that can utilize the patterns hidden in textual feedback. In our experiments, we find that the inclusion of such feedback greatly improves the efficiency and overall accuracy of our method, sometimes surpassing existing white-box methods (Zhou et al., 2022a; Wortsman et al., 2021) on a challenging one-shot scenario.

**Our contributions.** In this work, we introduce a novel method for black-box prompt engineering of VLMs, utilizing an LLM as an *optimizer*. Our gradient-free approach can surprisingly compete with various white-box methods in a low-shot setting. Additionally, we extensively explore various strategies for conversing with ChatGPT, uncovering several key factors that significantly enhance the efficiency of this tool. We also show that our discovered natural language prompts are not only *interpretable* but also *transfer* better across CLIP architectures, eg., from RN50 to ViT/B-16, than continuous prompts discovered by previous white-box prompting method (Zhou et al., 2022a).

## 2 RELATED WORK

**LLMs for multimodal tasks.** Cutting-edge LLMs like GPTs (Ouyang et al., 2022; OpenAI, 2023) have been successfully applied to multimodal tasks, either through zero-shot composition with pre-trained multimodal models (Li et al., 2022b; Zeng et al., 2022) or by jointly finetuning with modality-specific encoders (Li et al., 2023; Alayrac et al., 2022) on large-scale multimodal datasets (Schuhmann et al., 2022). LLMs are also utilized as neuro-symbolic reasoners (Gupta & Kembhavi, 2022; Shen et al., 2023; Lu et al., 2023; Zheng et al., 2023), translating natural language instructions into modular programs (like Python code) that invoke APIs of multimodal models. In this work, we show the potential of LLMs as a *black-box optimizier* for multimodal foundation models with language interfaces, and more specifically vision-language models (VLMs).

**Black-box optimization of foundation models.** Owing to privacy and legal constraints (Tramèr et al., 2016; Xiao et al., 2023), foundation models (Brown et al., 2020; OpenAI, 2023) are commonly provided as cloud services accessible only via API calls, making popular fine-tuning approaches such as LoRA (Hu et al., 2021) and adapters (Houlsby et al., 2019) infeasible without model parameters and gradients. Following the success of in-context learning (Brown et al., 2020), which appends user-generated natural language instruction and few-shot samples to text inputs, prompting (Liu et al., 2023) has emerged as the preferred finetuning paradigm for LLMs due to its superior performance and parameter-efficiency. While most prompting methods operate in a white-box manner, such as continuous prefix-tuning (Li & Liang, 2021) or discrete token-searching (Shin et al., 2020), black-box prompting methods have recently emerged. Because the numerical computation of a black-box gradient via finite-differencing is impractical, contemporary black-box methods tend to employ (a) heuristic-based editing (Prasad et al., 2022; Mishra et al., 2021), (b) continuous prefix-tuning with genetic algorithms (Sun et al., 2022b; Xu et al., 2022; Chai et al., 2022; Sun et al., 2022a), and (c) discrete token searching with reinforcement learning (Deng et al., 2022; Diao et al., 2022). Despite their novelty, these methods face challenges with efficiency, large variance, interpretability, and hyperparameter sensitivity (Hou et al., 2022). Instead, we propose to leverage state-of-the-art LLMs, such as ChatGPT (Ouyang et al., 2022; OpenAI, 2023), to iteratively optimize prompts via *conversational* feedback. Our method can discover a single interpretable prompt for VLMs, and outperform existing work such as Menon & Vondrick (2022) that performs prompt ensembling by averaging the output logits.

**LLMs for prompt optimization.** LLMs have proven to be an effective external knowledge base (Shen et al., 2022; Menon & Vondrick, 2022; Pratt et al., 2022) for generating prompts in a zero-shot setting for multimodal models. For example, DCLIP (Menon & Vondrick, 2022) uses GPT3 to come up with rich visual descriptions to improve zero-shot classification with CLIP (Radford et al., 2021). We extend this line of work to show that LLMs can *iteratively* optimize prompts for VLMs in a black-box fashion given a few-shot validation set. Closet to our work is APE (Zhou et al., 2022b), which leverages an LLM to write prompts for another LLM using few-shot samples based on instruction induction (Honovich et al., 2022) and iterative Monte Carle search, ie., paraphrasing the current prompt in a hill-climbing fashion (Russell, 2010; Mitchell et al., 1993). However, APE is designed to address language tasks, while we focus on multimodal tasks using black-box VLMs. We further illustrate that prompt optimization with LLMs can be made more efficient by leveraging *conversational* feedback, such as providing ChatGPT with explicit language feedback on how well

---

**Algorithm 1** General prompt engineering framework. This algorithm shows how humans perform prompt engineering, which motivates our method of prompt engineering using chat-based LLMs.

---

**Require:** $D_{\text{train}} = \{x, y\}_n$: training samples, $F : D \times T \to \mathbb{R}$: black-box score function
 1: Create an initial prompt set: $\mathcal{U} \leftarrow \{p_1\}$
 2: Evaluate the initial prompt on training set: $S \leftarrow \{F(D_{\text{train}}, p_1)\}$
 3: **while** not converged **do**
 4:     Generate a new prompt $p'$ based on $S$
 5:     Evaluate the score of the new prompt on few-shot samples: $s' = F(D_{\text{train}}, p')$
 6:     $\mathcal{U} \leftarrow \mathcal{U} \cup \{p'\}$
 7:     $S \leftarrow S \cup \{s'\}$
 8: **end while**
 9: **return** prompt with highest score $p^* \leftarrow \arg\max_{p \in \mathcal{U}} F(D_{\text{train}}, p)$

---

the most recent prompt performs. Our findings align with the perspective (Dai et al., 2022) of LLMs as meta-optimiziers that can implicitly perform gradient search through in-context learning.

**Few-shot adaptation of VLMs.** Prompting has also been successfully adopted in VLMs (Gan et al., 2022), as demonstrated by methods like CoOp (Zhou et al., 2022a) that finetune an ensemble of continuous prefix tokens using cross-entropy loss. Lin et al. (2023) achieves state-of-the-art few-shot performance with a cross-modal (image and text) cross-entropy loss. However, these methods all require access to model parameters and outputs for gradient backpropagation. In this work, we introduce a gradient-free approach based on chat-based LLMs, yielding competitive results to white-box approaches in extremely low-shot scenarios.

## 3 CONVERSATIONAL PROMPTING USING CHAT-BASED LLMS

We now present our approach for **conversational prompting** using chat-based LLMs as optimizers.

**Preliminaries.** We assume that the targeted task is accompanied by a training dataset denoted as $D_{train} = \{x, y\}_n \subset D$. Our objective is to enhance the performance of a black-box model equipped with a language interface capable of processing a textual prompt $p \in T$. We assume a *black-box score function* represented as $F : D \times T \to \mathbb{R}$. This function leverages the black-box model to compute a score $F(D_{train}, p)$ which indicates the performance of the textual prompt on this dataset, such as classification accuracy. It is noteworthy that our setup requires *minimal* knowledge about the black-box model, which is crucial since even releasing output logits or embeddings can potentially facilitate the unauthorized extraction of knowledge through white-box distillation (Hinton et al., 2015). Our goal of prompt engineering is to search for the optimal *natural language prompt* $p^*$ without accessing or modifying the black-box model.

**Background: human prompt engineering for black-box models.** Our method is inspired by the typical workflow of human prompt engineers. Prompt engineering is often an iterative process that involves: (a) creating an initial prompt $U = \{p_1\}$ based on understanding of the task, (b) testing prompts in $U$ using the black-box scoring function, (c) refining prompts based on the outcomes, (d) repeating the last two steps until convergence, and (e) returning the prompt $p^*$ with highest $F(D_{train}, p^*)$. This hands-on approach helps to optimize the model's performance for specific tasks, but can be tedious and labor-intensive. Algorithm 1 formally illustrates this process.

**Example: manual prompting for image classification with CLIP (Radford et al., 2021).** CLIP is one of the most popular VLM that takes a set of class-specific prompts when performing "zero-shot" image classification. Radford et al. (2021) details the laborious prompting procedure over the course of a year. Interestingly, they find that a default class-agnostic prompt (or so-called "template"), ``a photo of a {class}'' can provide a decent boost in accuracy for most datasets compared to using vanilla class labels. In this scenario, the score function $F$ is the classification accuracy on the test set, and the prompt $p = \{$``a photo of a {c}''$|c \in C\}$, where $C$ is the set of class names for a given dataset.

**Leveraging LLMs as prompt engineers (prior art).** Recent studies such as *Automatic Prompt Engineer* (Zhou et al., 2022b) show that LLMs are on par with human prompt engineers across various language tasks. Specifically, they apply in-context learning by providing an LLM such as

---

**Algorithm 2** Black-box prompt engineering framework through conversational prompting using LLMs. Our algorithm works with any general setup involving a chat-based LLM and a black-box score function, such as accuracy for classification and mAP for retrieval. We highlight mechanisms for "exploration" (restart and reset) in blue and "exploitation" (iter) in red. We mark the key component of "conversational feedback" of our approach in violet. The actual prompts are attached in section 7.

---

**Require:** $D_{\text{train}} = \{x, y\}_n$: training samples, $F : D \times T \to \mathbb{R}$: black-box score function.
**Require:** $n_{\text{restart}}$: number of initial sampled prompt sets, $n_{\text{reset}}$: number of resets for a prompt set, $n_{\text{iter}}$: number of hill-climbing iterations, $m$: size of one initial prompt set, $k$: number of prompt samples send to ChatGPT.

1:   $p^* \leftarrow \emptyset$
2:   **for** $1::n_{\text{restart}}$ **do**
3:      Sample a new prompt set from a text corpus, $\mathcal{U}_{\text{init}} \leftarrow \{p_1, ..., p_m\}$
4:      **for** $1::n_{\text{reset}}$ **do**
5:         Reset to initial prompt set: $\mathcal{U} \leftarrow \mathcal{U}_{\text{init}}$
6:         **for** $1::n_{\text{iter}}$ **do**
7:            Sort $\mathcal{U}$ based on their scores on training samples using $\{F(D_{\text{train}}, p)\}_{p \in U}$
8:            $\mathcal{U}_{\text{top}} \leftarrow$ top-k prompts in $\mathcal{U}$
9:            $\mathcal{U}_{\text{bot}} \leftarrow$ bottom-k prompts in $\mathcal{U}$
10:          Generate a new prompt based on top and bottom-k prompts $p_{\text{new}} \leftarrow \text{LLM}(\mathcal{U}_{\text{top}}, \mathcal{U}_{\text{bot}})$
11:          $\mathcal{U} \leftarrow \mathcal{U} \cup \{p_{\text{new}}\}$
12:         **end for**
13:         Update best prompt: $p^* \leftarrow \arg\max_{p \in \mathcal{U} \cup \{p^*\}} F(D_{\text{train}}, p)$
14:      **end for**
15: **end for**
16: **return** prompt with highest score $p^*$

---

GPT-3 (Brown et al., 2020) a few examples of (input, output) pairs, and ask it to reverse-engineer the task instruction. Moreover, they propose an *iterative Monte Carlo search* to continuously improve the instruction by asking GPT-3 to *paraphrase* the best performing one so far. While reverse-engineering prompts from few-shot samples is only applicable to language tasks where task descriptions serve as prompts, we adopt a similar hill-climbing strategy as the prompt optimization framework.

**Conversational prompting with chat-based LLMs (our approach).** While APE primarily focuses on "instruction generation" through dataset samples and it is only applicable to language tasks, our approach delves deeper into the "optimization" aspect, envisioning the LLM as a *black-box optimizer* of prompts. Rather than merely asking a LLM to "blindly" augment existing candidate prompts, we make this process more efficient by *explicitly* providing **feedback** on the prompts, akin to how human prompt engineers refine prompts based on repetitive trials. Specifically, we maintain a pool of prompts $U$ and their corresponding scores $S$. In each iteration, we provide the LLM with both *positive* and *negative* prompts, such as the highest and lowest-performing candidates. Such textual feedback through in-context prompts offers LLMs an implied "gradient" direction (Dai et al., 2022), making optimization more efficient than taking random local steps. We facilitate this feedback mechanism through *conversations* with state-of-the-art chat-based LLMs like ChatGPT (Ouyang et al., 2022) as illustrated in Figure 1. We also note that the multi-turn conversation in Figure 1 is not the only way of conversing with ChatGPT and we ablate different forms of conversational feedback in section 7.

**Key features of our approach.** We outline our approach in Alg. 2. To start, since we cannot generate prompts directly from LLMs through instruction-induction (Zhou et al., 2022b) using (image, label) pairs, we opt to sample entirely random initial prompts from a text corpus such as LAION-COCO (Schuhmann et al., 2021) captions. Our approach follows the classical *stochastic hill-climbing framework with random-restart* (Russell, 2010), which prevents ChatGPT from being trapped in local optima by balancing "exploration" and "exploitation". Our **restart** mechanism is implemented by sampling $n_{\text{restart}}$ initial prompt sets to encourage exploration. Because ChatGPT performs stochastic top-k sampling for text generation (as we adopt the default temperature of 1.0), we also implement a **reset** mechanism to foster additional exploration by retrying a given prompt set $n_{\text{reset}}$ times. For exploitation, we converse with ChatGPT for $n_{\text{iter}}$ iterations. We find that it is critical to balance exploration and exploitation for optimal performance, and thoroughly examine this trade-off in section 5. Lastly, our **conversational prompting** iteratively provides textual feedback to

ChatGPT based on the top and bottom-performing prompts, denoted as $(\mathcal{U}_{top}, \mathcal{U}_{bot})$. We show that this simple adjustment is critical to the efficiency of our approach in Figure 2.

## 4  ILLUSTRATIVE TASK: FEW-SHOT IMAGE CLASSIFICATION

We illustrate our approach using a few-shot image classification task. Specifically, a prompt $p \in T$ consists of a set of class-specific prompts – that is, one textual description per class. The scoring function $F$ takes the prompt $p$, along with an image dataset $D_{train}$, and returns the average per-class accuracy using the black-box VLM. To prevent overfitting and simplify our search space, we restrict our search to finding a single class-agnostic template, e.g., `a photo of a {}`, filling in the blank with label names provided with the dataset.

**Experimental setup.** We apply our approach to the few-shot image classification benchmark introduced in CoOp (Zhou et al., 2022a), which is the most commonly studied setup for fine-tuning of VLMs. This benchmark involves a collection of 11 datasets covering diverse image domains including ImageNet (Deng et al., 2009) and more niche datasets such as FGVC-Aircraft (Maji et al., 2013). For each dataset, we adhere to the same three-fold k-shot train sets in (Lin et al., 2023), reporting the average accuracy across all folds. Importantly, our method only utilizes the train set to compute the score and does not require the few-shot validation set. We use CLIP following prior work (Lin et al., 2023; Zhou et al., 2022a) to emulate a black-box VLM, and we employ ChatGPT (GPT3.5) as the LLM to facilitate conversational prompting.

**Implementation details.** We randomly sample 1M captions from the LAION-COCO (Schuhmann et al., 2021). For each caption, we extract all the noun phrases using spaCy part-of-speech tagging (Honnibal & Montani, 2017). Subsequently, we replace one noun phrase in the caption with `` `{}' `` (a placeholder where the class name will be inserted) to create a template. Given that each caption contains an average of 2 noun phrases, our initial prompt pool consists of approximately 2M templates. We run our algorithm with $n_{restart} = 20$ restarts, $n_{restart} = 50$ resets, and $n_{restart} = 10$ iterations. We opt to sample $m = 100$ prompts per restart and present the top and bottom $k = 15$ prompts to ChatGPT. We ablate different sets of hyperparameter and explain how we balance the tradeoff between exploration and exploitation in section 8. We adopt `gpt-3.5-turbo-0301` model for ChatGPT using OpenAI's official API and keep the default sampling temperature of 1.0. We also ablate `gpt-4` in Table 7 and find it achieve similar performance. The exact prompts used to converse with ChatGPT are documented in section 7. For a fair comparison, we use CLIP-RN50 for our experiments following prior work (Lin et al., 2023; Zhou et al., 2022a). We will open-source our code and release the initial prompt pool (LAIONCOCO-1M) to the public.

**Previous white-box baselines.** Our black-box setup substantially differs from, and is more constrained than, the scenarios considered in previous white-box baselines. Specifically, we do **not** expose the pre-trained weights, model architectures, feature embeddings, or even output logits of VLMs. These constraints render many established *gradient-based fine-tuning* baselines inapplicable. Among the white-box approaches we later compare to, **CoOp** (Zhou et al., 2022a) performs continuous prompting and requires backpropagation across all layers. **WiSE-FT** (Wortsman et al., 2021) ensembles fine-tuned weights with the original CLIP weights. **Cross-Modal Adaptation** (Lin et al., 2023) fine-tunes a linear classifier leveraging both image and text embeddings from CLIP. Finally, while **DCLIP (Menon & Vondrick, 2022)** queries GPT3 for rich visual descriptors for each class and does not require gradient-based finetuning, it performs *prompt ensembling* using 4-6 class-specific prompts per class, which breaches our black-box assumption for accessing the output logits.

**Black-box methods.** We additionally benchmark our method against truly black-box solutions, including the vanilla class-agnostic templates `` `{classname}' `` and `` `a photo of a {classname}' ``. Also, we compare our approach to the best **Hand-Engineered** templates released by OpenAI, searched using *test set* performance to represent the theoretical upper bound of human performance, e.g., `` `a centered satellite photo of {classname}.' `` for EuroSAT (Helber et al., 2017). Finally, we present two versions of conversational feedback of our approach: (a) using 30 positive (**P only**) or (b) using 15 positive and 15 negative prompts (**P+N**) in each iteration. For a fair comparison, both of our approaches start with the same initial sampled prompts, referred to as **LAIONCOCO-1M**. We also show the performance of the best initial sampled prompt searched using trainset performance.

| Method | BB | Dataset | | | | | | | | | | | Avg |
|--------|----|---------|---------|----------|------|------|------|------|------|------|--------|---------|-----|
| | | Caltech | ImageNet | Aircraft | Food | Pets | Cars | SUN | UCF | DTD | EuroSAT | Flowers | |
| Cross-Modal | ✗ | 89.1 | 61.6 | 20.6 | 77.1 | 85.7 | 59.0 | 63.4 | 64.7 | 49.9 | 61.8 | 76.3 | 64.7 |
| Wise-FT | ✗ | 85.5 | 58.3 | 18.6 | 71.9 | 81.7 | 55.7 | 56.6 | 59.4 | 44.2 | 52.3 | 65.8 | 59.1 |
| CoOp | ✗ | 87.5 | 57.2 | 9.6 | 74.3 | 85.9 | 55.6 | 60.3 | 61.9 | 44.4 | 50.6 | 68.1 | 59.6 |
| DCLIP | ✗ | - | 59.6 | - | 76.4 | 83.8 | - | - | - | 41.7 | 34.7 | - | - |
| {} | ✓ | 78.5 | 55.3 | 15.5 | 74.0 | 78.9 | 52.2 | 53.4 | 55.5 | 41.4 | 32.1 | 57.3 | 54.0 |
| a photo of a {} | ✓ | 84.5 | 57.9 | 15.9 | 74.0 | 83.2 | 53.9 | 58.0 | 56.9 | 38.8 | 28.6 | 60.2 | 55.6 |
| Hand-Engineered | ✓ | 86.3 | 58.2 | 17.3 | 77.3 | 85.8 | 55.6 | 58.5 | **61.5** | 42.3 | 37.6 | 66.1 | 58.8 |
| LAIONCOCO-1M | ✓ | 81.4 | 56.2 | 17.4 | 76.5 | 79.6 | 51.3 | 54.9 | 55.8 | 43.1 | 38.6 | 61.3 | 56.0 |
| Ours (P only) | ✓ | 89.0 | 59.4 | 17.9 | 77.8 | 85.7 | 55.7 | 60.4 | 58.7 | 43.6 | 46.7 | 66.6 | 60.1 |
| Ours (P+N) | ✓ | **89.1** | **59.6** | **18.1** | **78.3** | **88.1** | **56.2** | **61.0** | 60.2 | **44.8** | **49.0** | **67.2** | **61.1** |

Table 1: **Comparison of conversational prompting (our method) with other baselines on one-shot classification tasks.** We report the average accuracy of each method across three folds, optimized using 1-shot training sets. We mark all white-box solutions in gray, as they require either gradient-based fine-tuning (CoOp/WiSE-FT/Cross-Modal) or prompt ensembling using output logits (DCLIP). We **bold** the best black-box (**BB**) result for each dataset, and underline the second best result. First, we note that our approach can effectively improve upon the initial prompts selected from LAIONCOCO-1M from $56\%$ to $61\%$. Our approach is also competitive against the best Human-Engineered prompts released by OpenAI searched using *test set* performance. Remarkably, we also surpass white-box solutions such as WiSE-FT and CoOp by at least $1.5\%$. Finally, we show that using both positive and negative prompts improves the overall accuracy by $1\%$. While our approach is less effective than the SOTA white-box method (Cross-Modal Adaptation), we stress that our black-box setup is significantly more challenging, because we restrict the optimization space to *natural language* and do *not* access the pre-trained weights, model architectures, feature embeddings, and output logits of VLMs.

**SOTA one-shot performance against existing methods on 11 datasets.** We report the test set performance of conversational prompting (**Our Method**) versus the aforementioned baselines in a challenging 1-shot classification scenario in Table 1. First, compared to the top-performing initial prompts selected from **LAIONCOCO-1M** based on train set performance, our prompt optimization using ChatGPT notably improves the initial prompts by an average of $5\%$ ($56\%$ to $61\%$). Remarkably, our black-box approach surpasses the two white-box gradient-based fine-tuning techniques CoOp and WiSE-FT by at least $1.5\%$. Given that both CoOp and our method optimize a single class-agnostic template, we attribute this gap in performance to *reduced overfitting*. More specifically, we posit that our optimization space of natural language effectively acts as a regularizer in extremely low-shot tasks, standing as a more robust alternative to the continuous prompting approach of CoOp. Furthermore, our method benefits from textual feedback and shows improved performance by $1.0\%$ when using both positive and negative prompts. In section 8, we show that our approach remains effective across different CLIP and ChatGPT variants.

**Incorporating negative prompts leads to more efficient optimization.** In Figure 2, we demonstrate that incorporating **both positive and negative prompts** fosters better optimization efficiency, achieving higher accuracy within a much fewer number of resets. Specifically, we hypothesize that LLMs can leverage the implicit "gradient" direction suggested in textual feedback to achieve faster convergence. For additional analysis, we ablate different ways of providing conversational feedback to ChatGPT in section 7 and conclude that iteratively updating both positive and negative prompts is the key for efficient optimization.

## 5 ADDITIONAL BENEFITS OF NATURAL LANGUAGE PROMPTS

In this section, we delve deeper into the advantages of utilizing natural language prompts compared to the continuous prompts (Zhou et al., 2022a). We specifically highlight that the prompts derived through our method are *interpretable*; for instance, they often contain descriptions of the targeted image domain. We also illustrate that these prompts maintain a higher degree of *transferability* across varying CLIP architectures in a *black-box manner*, such as transferring from RN50 to ViT/B-16.

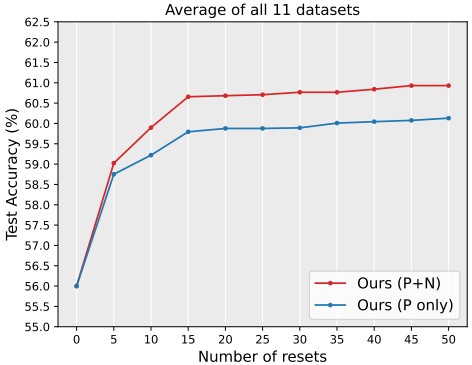 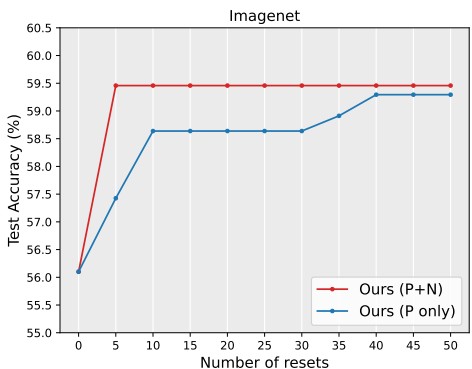

Figure 2: **Conversational feedback incorporating both positive and negative prompts leads to improved efficiency.** We fix the number of restarts to 20 and iterations to 10, and ablate different numbers of resets on all 11 datasets (left) and ImageNet (right). Notably, our approach using "P+N" (both top-k and bottom-k prompts) can optimize faster within a much fewer number of resets than using "P-Only" (top-2k prompts), resulting in the highest overall performance.

| Dataset | Example of Top Templates |
|---|---|
| Caltech | An image of a {} with a blurred background that emphasizes the subject |
| DTD | The essential elements of {} are amplified with visual simplicity |
| EuroSAT | A top-down view of {} arranged in a pattern {} |
| Aircraft | A clear, high-quality image of a single {} with a white background |
| Food | A {} featuring diverse cuisine and ingredients |
| ImageNet | An image of a {} with bright and natural lighting |
| Flowers | A clear and vivid photograph of the {} in its natural setting |
| Pets | A {} with distinct and recognizable characteristics |
| Cars | A {} featuring a wide range of color options for easy selection |
| SUN | A high-resolution photo of a {} with clear background and natural lighting |
| UCF | A black and white photo of a {} in motion |

Table 2: **Example templates returned by our algorithm on each dataset.** Although we do not provide ChatGPT with any information regarding the targeted dataset, we observe that the resulting templates are remarkably similar to human-engineered templates, with many domain-specific details such as "motion" and "cuisine", and stylistic elements such as "bright and natural lighting".

**Our method discovers interpretable natural language prompts.** While CoOp (Zhou et al., 2022a) concedes that continuous prompts can be difficult to interpret, even with nearest-neighbor searches to find the closest discrete tokens, our method – without explicitly instructing ChatGPT to do so – often yields interpretable results. Table 2 showcases the templates returned by our algorithm for each dataset, frequently including keywords that reflect the targeted image domain. For example, the template for Food101 (Bossard et al., 2014) mentions "diverse cuisine and ingredients", and the template for UCF101 (Soomro et al., 2012) (an action recognition dataset) mentions "in motion". Likewise, these templates identify general stylistic attributes of the datasets; they refer to "bright and natural lighting" for ImageNet (Deng et al., 2009) and note images that "emphasize the subject" for Caltech101 (Li et al., 2022a). These prompts are particularly intriguing because we do not provide ChatGPT with any information about the downstream task, yet it manages to generate prompts containing domain-specific keywords that are similar to those engineered by human experts.

**Prompt transferring across CLIP architectures in a black-box manner.** The natural language prompts discovered through our approach also exhibit strong transferability to different CLIP backbones, maintaining consistently high performance. For comparison, since CoOp uses the same tokenizer for all CLIP architectures (including ResNet-50, ResNet-101, ViT/B-32, and ViT/B-16) and optimizes continuous prompts of the same shape (16 x 512), we assess the transferability of these learned continuous prompts from RN50 to other backbones using the official weights on 16-shot Ima-

| Method | RN50 | →RN101 | →ViT-B/32 | →ViT-B/16 |
|---|---|---|---|---|
| a photo of a {} | 57.9 | 60.6 | 61.9 | 66.6 |
| CoOp | **63.0** | 20.6 | 31.7 | 39.5 |
| Ours | 59.9 | **60.7** | **62.2** | **67.0** |

Table 3: **Black-box transferring prompts from ResNet-50 to other CLIP architectures on 16-shot ImageNet.** We evaluate both our natural language prompts and CoOp's continuous prompts, which are trained using the RN50 CLIP backbone on a 16-shot ImageNet setup, by reporting their test performances across different CLIP backbones. We choose to report the 16-shot results because those are the only publicly available CoOp weights. As a reference point, we include the baseline prompt "a photo of a ", and show that the prompts derived from our method using RN50 consistently surpass it after transferring to different backbones. In contrast, while CoOp achieves better 16-shot ImageNet performance using RN50, it exhibits a significant decline; for instance, its performance plummets from 63% to a mere 21% during the RN101 transfer. This stark difference highlights that our approach effectively generates more generalizable natural language prompts, thereby avoiding the overfitting issue associated with the specific CLIP architecture.

geNet. Table 3 showcases the results of this experiment, where we also include the baseline prompt a photo of a {} for reference. We observe a significant decline in accuracy when transferring CoOp's prompts (up to a 40% decrease despite utilizing more powerful backbones), implying that continuous prompts tend to overfit to the specific CLIP model. In contrast, our natural language prompts maintain their performance and outperform the baseline prompt across all backbones.

## 6 DISCUSSION AND LIMITATIONS

**Summary.** We present the first attempt to leverage LLMs as prompt engineers for black-box VLMs, mirroring the iterative process humans typically undertake to engineer prompts. On the well-studied setup of one-shot image classification, our method surpasses existing human-engineered prompts and even rivals white-box approaches. Central to the success of our method is the utilization of both positive and negative prompts, enabling chat-based LLMs to efficiently steer VLMs in the right direction using the textual feedback. This process leads to a set of interpretable prompts bearing considerable resemblance to those crafted by humans. Importantly, our natural language prompting setup is a lot more constrained than the assumed scenarios of previous white-box approaches, because we do not expose the mode weights and output embeddings of VLMs. While this setup is arguably challenging, our method learns effective and interpretable prompts that generalize across black-box VLMs.

**Limitations and future work.** As with any work utilizing LLMs, there are various ethical concerns, including the potential for biases in the LLM's output. Such biases could propagate through the LLM and affect the final prompt output. Moreover, while we try to minimize the overall cost and the total number of API calls/tokens used, the energy consumption associated with LLMs remains a substantial concern. We emphasize that this work is an initial exploration of conversational prompting with LLMs and is not intended for real-world application at this stage. It is vital to note that we employ white-box baselines merely as a reference in Table 1, without intending to compete directly with white-box methods. Indeed, our approach lags behind the leading white-box solution, **Cross-Modal Adaptation (Lin et al., 2023)** that utilize embeddings, logits, and gradients to improve visual and text representations with more data. Further details on the higher-shot performance of our method can be found in section 8. Looking forward, we aspire to inspire further exploration into leveraging LLMs as conversational optimization tools. Future work may extend our method to other multimodal tasks, such as text-to-image generation.

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

## 7 DETAILS OF CONVERSING WITH CHATGPT

**Multi-turn conversation.** We use ChatGPT to generate a set of new prompts based on the top and bottom performing prompts (line 10 of Algorithm 2). The exact prompts we use are:

```
Hi ChatGPT, assume you are a pattern learner.  I have two
lists of CLIP templates:  one with good templates and the
other with bad templates.  There are latent patterns that make
a template good or bad.  Based on these patterns, give me a
better template for image classification while avoiding worse
template.
Here is the list of good templates:
- good1
- good2
- ...
Here is the list of bad templates:
- bad1
- bad2
- ...
Here are my requirements:
- Please only reply with the template.
- The template should be fewer than 15 words.
- The template should have a similar structure to the above
templates.

Positive Response (if the new prompt outperforms the top-k)
The performance of the template ``newTemplate'' improves to
X.XX%.  Please give me a better template.

Negative Response
The performance of the template ``newTemplate'' drops to X.XX%.
Please give me a better template.
```

**Alternative implementation: sending only the initial prompts (default).** Multi-turn conversation requires appending all chat history to ChatGPT's official API at every iteration, which costs more input tokens and money. In Figure 3, we show that one can only send the initial prompts (without any response) to ChatGPT at every iteration to get equivalent and even slightly better performance. However, it is important to also update the top-k and bottom-k prompts at every iteration (**Iterative**) for efficiency. We show that the **Non-Iterative** version that keeps re-using the initial top-k and bottom-k prompts leads to worse performance. Therefore, in our paper, we stick to **Iterative** for all experiments.

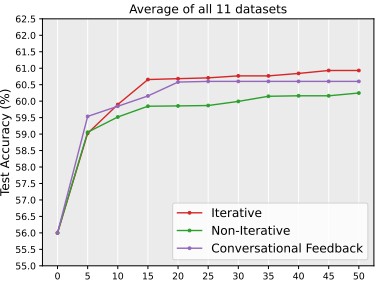 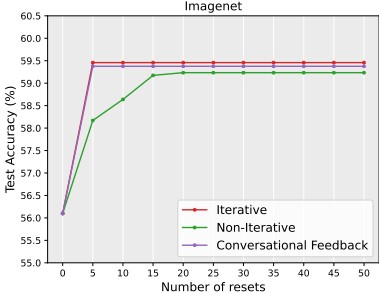

Figure 3: **Updating initial prompts can be as effective as multi-turn conversation.** We ablate different ways of conversing with ChatGPT on all 11 datasets (left) and ImageNet (right). Notably, we find that only updating the top-k and bottom-k prompts (**Iterative**) is as performant and thus a cheaper alternative because sending response to ChatGPT costs more input tokens. On the other hand, reusing the initial prompts (**Non-Iterative**) leads to worse overall performance.

**Positive Only (P only).** When using only positive prompts, we can remove negative prompts and provide twice as many positive examples:

```
Hi ChatGPT, assume you are a pattern learner.  I have one list
of CLIP templates:  one with good templates.  There are latent
patterns that make a template good.  Based on these patterns,
give me a better template for image classification.
Here is the list of good templates:
- good1
- good2
- ...
Here are my requirements:
- Please only reply with the template.
- The template should be fewer than 15 words.
- The template should have a similar structure to the above
templates.
```

## 8 ADDITIONAL EXPERIMENTAL RESULTS

In this section, we present additional experiments to gain further insights into our method.

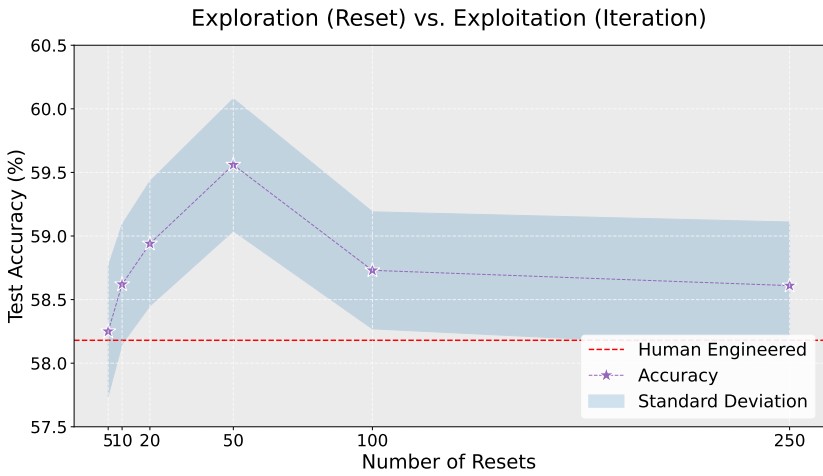

Figure 4: **Balancing exploration and exploitation.** We use a fixed budget of 500 ChatGPT API calls per restart, and ablate the optimal number of resets to use in our algorithm on 1-shot ImageNet. The number of iterations is thus inversely proportional to the number of resets; for example, 10 resets would allow for 50 iterations per reset. We take the average over three runs and also report the standard deviation. We find the optimal balance of exploration and exploitation to be 10 iterations and 50 resets. In contrast, "pure" exploration (2 iterations, 250 resets) leads to 0.9% lower accuracy due to insufficient optimization. On the other hand, when exploitation is overly prioritized (100 iterations, 5 resets), our method gets 1.3% lower accuracy.

**Balancing exploration and exploitation can improve the final performance.** Our method extensively leverages the ChatGPT API, necessitating an investigation into strategies for minimizing optimization costs. This leads us to examine the classic dilemma of exploration versus exploitation, a foundational concept in reinforcement learning. Specifically, we use a fixed budget of 500 API calls per restart, and investigate the optimal combination of the number of resets and iterations in Figure 4. For example, we can allocate 50 resets with 10 iterations each to encourage more exploration, or 10 resets with 50 iterations each to foster more exploitation. We find that the optimal balance point is 50 resets of 10 iterations each, and note that no other combination is within 1 standard deviation of the optimal performance. As shown in the performance curve, having too much exploration (250 resets), or too little (5 resets) will result in a roughly $1\%$ decrease in performance. In general, we find

| Backbone | Method | | |
|---|---|---|---|
| | Our Approach | Hand-Engineered | Linear Probe |
| ResNet-50 | 59.6 | 58.2 | 55.9 |
| ResNet-101 | 61.8 | 61.6 | 59.8 |
| ViT-B/32 | 62.6 | 62.0 | 59.6 |
| ViT-B/16 | 67.8 | 66.7 | 65.9 |

Table 4: **Our method can generalize to various CLIP architectures.** We run our method on 1-shot ImageNet across multiple CLIP backbones, and compare it to the best Human-Engineered prompt and Linear-Probing (Radford et al., 2021) performance.

| Method | BB | Dataset | | | | | | | | | | | Avg |
|---|---|---|---|---|---|---|---|---|---|---|---|---|---|
| | | Caltech | ImageNet | Aircraft | Food | Pets | Cars | SUN | UCF | DTD | EuroSAT | Flowers | |
| LAIONCOCO-1M | ✓ | 81.4 | 56.2 | 17.4 | 76.5 | 79.6 | 51.3 | 54.9 | 55.8 | 43.1 | 38.6 | 61.3 | 56.0 |
| Iterative APE | ✓ | 88.3 | 58.1 | 17.0 | 77.3 | 85.1 | 54.8 | 58.6 | 57.4 | 41.2 | _46.7_ | 65.3 | 59.0 |
| Ours (P only) | ✓ | _89.0_ | _59.4_ | _17.9_ | _77.8_ | _85.7_ | _55.7_ | _60.4_ | _58.7_ | _43.6_ | _46.7_ | _66.6_ | _60.1_ |
| Ours (P+N) | ✓ | **89.1** | **59.6** | **18.1** | **78.3** | **88.1** | **56.2** | **61.0** | **60.2** | **44.8** | **49.0** | **67.2** | **61.1** |

Table 5: **Comparing our method with our own version of iterative APE (Zhou et al., 2022b).** Optimized using 1-shot training sets, we find that both iterative APE and our methods can effectively improve upon the initial sampled prompts. However, our method achieves better performance within the same computational budget, presumably because we provide explicit textual feedback to ChatGPT, leading to faster convergence.

it is useful to spend more budget on exploration as ChatGPT can be stuck at local minima within one reset.

**Using (iterative) APE for VLM optimization.** We attempt to implement our own version of iterative APE using the given prompts in Zhou et al. (2022b) while making minimal changes such that it fits in our automatic prompt searching system. For a fair comparison, we reuse exactly the same initial sampled prompts from LAIONCOCO-1M for iterative APE because their "instruction-induction" paradigm cannot be applied to VLM optimization settings. The results are shown in Table 5. We find that iterative APE shows inferior performance to our method, presumably because we leverage more textual feedback for more efficient search. The exact prompt we use is shown below:

```
Hi ChatGPT, generate a single variation of the following
template while keeping the semantic meaning:
- template
Here is my requirement:
- Please return a single template starting with '-'
```

**Comparison of CLIP backbones.** To verify that our method scales properly to other CLIP backbones, we test our method on ImageNet using four different CLIP backbones: ResNet-50, ResNet-101, ViT-B/32, and ViT-B/16. We compare our method with hand-engineered prompts, and a linear probe (linear classification on the visual embeddings). Table 4 shows the results of the experiment, where we see that our method outperforms the baselines consistently. Thus, our method scales appropriately with larger and more powerful models.

**Results on higher shots.** We additionally test the generalization ability of our method given more data (4 and 16 shots), with results shown in Table 6. We observe that our method gains small but incremental improvements given more data, and using both top-k and bottom-k prompts (P+N) consistently outperforms top-2k prompts (P only). We note that in the higher shot settings, our method cannot beat the performance of CoOp, but this is an unfair comparison because white-box methods such as CoOp can optimize over the continuous text-embedding space (a 16x512 size matrix) using actual gradients computed with cross-entropy loss. Therefore, it should not be surprising that these carefully crafted white-box fine-tuning methods can outperform us with higher shots. Given

| Shot | Method | Dataset | | | | | | | | | | | Avg |
|------|--------|---------|----------|----------|------|------|------|------|------|------|--------|---------|------|
| | | Caltech | ImageNet | Aircraft | Food | Pets | Cars | SUN | UCF | DTD | EuroSAT | Flowers | |
| 1 shot | Coop | 87.5 | 57.2 | 9.6 | 74.3 | 85.9 | 55.6 | 60.3 | 61.9 | 44.4 | 50.6 | 68.1 | 60.0 |
| | Ours (P only) | 89.0 | 59.4 | 17.9 | 77.8 | 87.8 | 55.7 | 60.4 | 58.7 | 43.6 | 46.7 | 66.6 | 60.1 |
| | Ours (P+N) | 89.1 | 59.6 | 18.1 | 78.3 | 88.1 | 56.2 | 61.0 | 60.2 | 44.8 | 49.0 | 67.2 | 61.1 |
| 4 shot | Coop | 89.6 | 60.0 | 21.9 | 73.3 | 86.7 | 62.6 | 63.5 | 67.0 | 53.5 | 70.2 | 86.2 | 66.8 |
| | Ours (P only) | 89.1 | 59.5 | 17.8 | 77.9 | 85.6 | 55.8 | 60.2 | 59.0 | 43.7 | 48.8 | 66.6 | 60.4 |
| | Ours (P+N) | 89.5 | 59.7 | 18.2 | 78.3 | 87.2 | 56.2 | 61.1 | 60.3 | 43.6 | 51.6 | 67.8 | 61.2 |
| 16 shot | Coop | 91.8 | 63.0 | 31.3 | 74.7 | 87.0 | 73.4 | 69.3 | 75.7 | 63.6 | 83.5 | 94.5 | 73.4 |
| | Ours (P only) | 89.3 | 59.6 | 17.7 | 77.9 | 86.6 | 56.2 | 61.0 | 60.2 | 44.0 | 49.0 | 66.0 | 60.6 |
| | Ours (P+N) | 89.5 | 59.9 | 18.1 | 78.3 | 88.3 | 56.8 | 60.8 | 60.5 | 44.9 | 51.4 | 67.4 | 61.4 |

Table 6: **Performance across 1/4/16 shots.** We report the higher-shot performance of both CoOp and our method in this table. The improvement on performance with more data is admittedly less significant than white-box methods such as CoOp, which can optimize over the continuous text-embedding space (a 30x512 size matrix) using actual model gradients. Still, our method's ability to match or even outperform such a white-box method in the 1-shot setting highlights its potential.

| GPT version | Dataset | | | | | | | | | | | Avg |
|-------------|---------|----------|----------|------|------|------|------|------|------|---------|---------|------|
| | Caltech | ImageNet | Aircraft | Food | Pets | Cars | SUN | UCF | DTD | EuroSAT | Flowers | |
| gpt-turbo-3.5-0301 | 89.1 | 59.6 | 18.1 | 78.3 | 88.1 | 56.2 | 61.0 | 60.2 | 44.8 | 49.0 | 67.2 | 61.1 |
| gpt-4-0314 | 89.1 | 59.6 | 17.9 | 78.5 | 87.7 | 56.2 | 60.3 | 59.9 | 45.0 | 48.0 | 67.6 | 60.9 |

Table 7: **ChatGPT versus GPT4.** Our approach is equally effective using other versions of ChatGPT.

sufficient labeled samples, these methods are more effective at fine-tuning better visual and text representations for downstream tasks.

**Results using GPT4.** We run our approach using the same hyperparameters and initial prompts using GPT4 in Table 7. It shows that our approach is equally effective using other versions of ChatGPT, but interestingly, there is no performance benefit of using GPT4. This may be because our hyperparameters were optimized on ChatGPT, and are suboptimal for GPT4.

**Cost analysis.** We use GPT3.5 which costs $0.0015 per 1000 tokens. In our default setup, we use an average of 500 tokens per API call. We use a total of 500 API calls (50 resets and 10 iterations) for a total of 250,000 tokens per restart, and thus each run costs around 50 cents. Since we use 20 restarts per dataset, the total cost over the suite of 11 datasets is around $100 for each of the three folds.

| GPT Model | Details | | | |
|-----------|-----------|--------------|-------------------------|-------------------------|
| | Tokens/Min | Requests/Min | Input Cost ($/1k tokens) | Input Cost ($/1k tokens) |
| gpt-turbo-3.5-0301 | 350,000 | 300,000 | $0.0015 | $0.002 |
| gpt-4-0314 | 40,000 | 200 | $0.03 | $0.06 |

Table 8: **Details of OpenAI API calls.** GPT4 is more costly and slower than GPT3.5.

