# OpenReview forum: "Language Models as Black-Box Optimizers for Vision-Language Models"
_ICLR.cc/2024/Conference — ICLR 2024 Conference Withdrawn Submission_

### Official Review · Reviewer_JF9P · 2023-10-30

**Soundness:** 3 good
**Presentation:** 3 good
**Contribution:** 2 fair
**Rating:** 5
**Confidence:** 4

**Summary:**

This paper attempts to leverage LLMs as prompt engineers for black-box VLMs. This method surpasses existing human-engineered prompts and even rivals white-box approaches on the well-studied setup of one-shot image classification. They use both positive and negative prompts, enabling chat-based LLMs to efficiently steer VLMs in the right direction using textual feedback. This process leads to a set of interpretable prompts bearing considerable resemblance to those crafted by humans. This method does not expose the mode weights and output embeddings of VLMs, so this setup is arguably challenging to learns effective and interpretable prompts that generalize across black-box VLMs.

**Strengths:**

1、Numerous existing vision-language models (VLMs) are not releasing their weights due to privacy and legal concerns, so it is difficult to employ current parameter-efficient tuning (PET) methods to adapt them to downstream datasets. Therefore, this paper proposes employing chat-based LLMs as black-box optimizers to search for the best text prompt on the illustrative task of few-shot image classification.

2、Technically, they adopt an automatic “hill-climbing” procedure (seems like an RL process) that converges on an effective prompt by evaluating the accuracy of current prompts and asking LLMs to refine them based on textual feedback, all within a conversational process without human-in-the-loop.

3、Although this paper focuses on prompt engineering, it distinguishes itself by offering a more interpretable approach compared to learnable prompt tuning methods.

**Weaknesses:**

1、The novelty of this paper is limited, as it shares similarities with APE[1] in utilizing LLMs to explore more efficient prompt templates although authors claim that their target is to optimize the prompt template for VLMs and adopt a similar hill-climbing strategy as the prompt optimization. This is more like an engineering improvement for optimizing prompt templates, not enough for a top-tier conference of ICLR that needs more explainable insights.

2、As illustrated in this paper, in the higher shot settings, this method cannot beat the performance of CoOp and other PET methods, rendering it impractical for most VLMs. Although authors argue this is an unfair comparison, in practice, PET methods that yield better performance, especially in generation tasks, are preferred.

3、While the paper claims the practicality of this method for existing VLMs that have not released their weights, it lacks experimental validation on these widely adopted VLMs, such as [2,3,4]. Besides, employing this method in generation tasks appears challenging due to the limited transferability of generation models using only text prompts.

[1]. Yongchao Zhou, Andrei Ioan Muresanu, Ziwen Han, Keiran Paster, Silviu Pitis, Harris Chan, and Jimmy Ba. Large language models are human-level prompt engineers. arXiv preprint arXiv:2211.01910, 2022b.

**Questions:**

1、While this method utilizes LLMs to optimize the prompt template, the stability, and the weak controllness in existing LLMs may limit the performance of this method. It would be beneficial to provide the variance of the experimental results and clarify this potential issue.

2、Conducting generalization experiments is necessary. For instance, investigating the effectiveness of utilizing the best template of ImageNet provided by LLMs on other datasets, such as "Oxford_pets," compared to the initial template in CLIP.

---

### Official Review · Reviewer_ipEa · 2023-10-31

**Soundness:** 3 good
**Presentation:** 3 good
**Contribution:** 2 fair
**Rating:** 5
**Confidence:** 4

**Summary:**

This paper proposes to use a language model (ChatGPT in their experiments) to generate prompts that optimize the classification performance of a vision-language model (CLIP in their experiments). The prompt optimization process mimics the prompt engineering method by humans. Specifically, some initial prompts are sampled from a prompt set to form a candidate prompt set and tested their performance with CLIP on a training set. The best 15 prompts and the worst 15 prompts are provided to ChatGPT, which is requested to generate a better prompt based on the input text containing the provided prompts. The generated prompt is tested on the performance of the training set and added to the candidate prompt set. This essentially provides feedback to ChatGPT on the effectiveness of generated prompts, thus allowing for automatic prompt engineering by ChatGPT. Experimental results on the one-shot setting with CLIP show improvement over some baselines, which includes the gradient-based prompt engineering method CoOp. Another experiment shows that the prompt obtained by this method is more generalizable to other Vision model backbones than CoOp.

**Strengths:**

- The proposed method shows improvement over the gradient-based prompt engineering method CoOp in the one-shot setting, which seems to suggest the proposed method is less prone to overfitting than gradient-based methods.
- The proposed method can have applications on extremely low-shot settings and/or settings where the scoring function is a black box.

**Weaknesses:**

- My major concern is on the applicability of this method. This method is probably suitable on extremely low-shot settings but once the number of training samples increases slightly, its advantage disappears quickly (e.g., with more than 4 shots). Moreover, most vision-language models are not black boxes and thus allow for gradient-based optimization. For this proposed method to work, it would be difficult to find applicable real scenarios where the vision-language model is a black box and there are only one or two training examples.

**Questions:**

- In the one-shot setting, how do the authors provide feedback to ChatGPT? There should be only one example in the training set. Is the performance still represented as X.XX%?

---

### Official Review · Reviewer_GEja · 2023-11-01

**Soundness:** 2 fair
**Presentation:** 3 good
**Contribution:** 2 fair
**Rating:** 5
**Confidence:** 4

**Summary:**

The paper proposes a conversational approach to query ChatGPT (or GPT-4) to iteratively modify the prompts for CLIP and reach stronger performance. The initial prompts (and the negative prompts) are collected from the LAION-COCO dataset, and ChatGPT will output the revised prompts based on the accuracy of the previous prompts. The results show that the approach can boost the performance of CLIP in the one-shot setting.

**Strengths:**

1. The paper utilizes ChatGPT to mitigate the manual annotation efforts.

2. The paper provides the estimated cost of the OpenAI APIs.

**Weaknesses:**

1. The paper didn't provide an experimental comparison to other black-box approaches, such as the methods mentioned in the related work (heuristic-based editing, continuous prefix-tuning, discrete token searching).

2. The approach only works better than Coop in the one-shot setting and does not scale well with the number of shots (Table 6), limiting the usefulness of the method.

--------

There are several typos "optimizier" on page 3 (maybe in other places too).

**Questions:**

1. How large is the training dataset to be used to estimate the accuracy, which is used as the feedback for the ChatGPT? If the size is large, then it means we need to query the "black-box" VLM (the paper's setup) frequently to get the accuracy, and I am not sure this is a realistic situation.

2. How does the performance of the prompt provided by the ChatGPT at each iteration change over the $n_{iter}$? If ChatGPT provides a better prompt based on the previous prompts and accuracy, I expect the accuracy should be somewhat monotonically increasing. Otherwise, the ChatGPT may just randomly guess the prompts and it may not learn anything from the past.

3. In Section 4, how to decide the top and bottom k prompts for the ChatGPT?

---

### Official Review · Reviewer_cyGF · 2023-11-01

**Soundness:** 3 good
**Presentation:** 4 excellent
**Contribution:** 1 poor
**Rating:** 3
**Confidence:** 5

**Summary:**

The authors use an LLM to propose language prompts that optimize performance on a CLIP-like model in a few-shot scenario. Since no gradients are involved, the method is black box (does not require access to the model weights). The LLM proposes language prompts (contrary to methods like coop that find prompts that optimizes prompt vectors in the continuous space so they might not correspond to words), thus being interpretable.

**Strengths:**

The paper is very well written and polished.

The method is very intuitive and easy to implement.

The results presented show some gains over CoOp

**Weaknesses:**

The method is very simple and the "our approach" part is a single paragraph. The idea is not very surprising as it is basically asking an LLM to do what a person would do manually, although it is interesting to see that it can surpass coop.

Comparisons are limited to the one-shot scenario. It is common in the prompting literature to provide results with 1, 2, 4, 8 and 16 shots (or at least the latter I believe?). This might look a bit like cherry-picking for maximal gains.

I understand that comparisons with white-box prompting methods are for reference, but there are no comparisons with black box ones. Aren't any of these methods from the past few conferences fair comparisons?
CVPR'23 BlackVIP: Black-Box Visual Prompting for Robust Transfer Learning
ICCV'23 Black Box Few-Shot Adaptation for Vision-Language models

**Questions:**

Responses to the weaknesses detailed above. It is not easy to reply to the novelty part (unless I misunderstood something), so clarifying experimental validation seems like the most promising.